# Isolation and Characterization of a Novel Hydrophobin, Sa-HFB1, with Antifungal Activity from an Alkaliphilic Fungus, *Sodiomyces alkalinus*

**DOI:** 10.3390/jof8070659

**Published:** 2022-06-23

**Authors:** Anastasia E. Kuvarina, Eugene A. Rogozhin, Maxim A. Sykonnikov, Alla V. Timofeeva, Marina V. Serebryakova, Natalia V. Fedorova, Lyudmila Y. Kokaeva, Tatiana A. Efimenko, Marina L. Georgieva, Vera S. Sadykova

**Affiliations:** 1Laboratory of Taxonomic Study and Collection of Cultures of Microorganisms, Gause Institute of New Antibiotics, St. Bolshaya Pirogovskaya, 11, 119021 Moscow, Russia; nastena.lysenko@mail.ru (A.E.K.); rea21@list.ru (E.A.R.); sukonnikoff.maxim@yandex.ru (M.A.S.); efimen@inbox.ru (T.A.E.); i-marina@yandex.ru (M.L.G.); 2Shemyakin and Ovchinnikov Institute of Bioorganic Chemistry, RAS, St. Miklukho-Maklaya, 16/10, 117997 Moscow, Russia; 3Belozersky Institute of Physico-Chemical Biology, Lomonosov Moscow State University, 119991 Moscow, Russia; 2.04.cochon@gmail.com (A.V.T.); mserebr@mail.ru (M.V.S.); fedorova@belozesky.msu.ru (N.V.F.); 4Faculty of Biology, Lomonosov Moscow State University, 1-12 Leninskie Gory, 119234 Moscow, Russia; kokaeval@gmail.com

**Keywords:** alkaliphilic fungi, antifungal peptide, *Sodiomyces alkalinus*, hydrophobin

## Abstract

The adaptations that alkaliphilic microorganisms have developed due to their extreme habitats promote the production of active natural compounds with the potential to control microorganisms, causing infections associated with healthcare. The primary purpose of this study was to isolate and identify a hydrophobin, Sa-HFB1, from an alkaliphilic fungus, *Sodiomyces alkalinus*. A potential antifungal effect against pathogenic and opportunistic fungi strains was determined. The MICs of Sa-HFB1 against opportunistic and clinical fungi ranged from 1 to 8 µg/mL and confirmed its higher activity against both non- and clinical isolates. The highest level of antifungal activity (MIC 1 µg/mL) was demonstrated for the clinical isolate *Cryptococcus neoformans* 297 m. The hydrophobin Sa-HFB1 may be partly responsible for the reported antifungal activity of *S. alkalinus*, and may serve as a potential source of lead compounds, meaning that it can be developed as an antifungal drug candidate.

## 1. Introduction

Extremophiles are a unique group of microorganisms capable of producing numerous bioactive compounds [1,2,3]. Alkaliphiles are extremophiles whose optimum rate of growth is at a high pH (9 to 11) [4,5]. Past decades of research have shown that the diversity of alkaliphilic fungi is less when compared to that of alkaliphilic bacteria. By now, the list of known alkaliphilic fungi (it is noteworthy that all are within the Ascomycota division) includes about 30 species [5,6,7]. Members of the genus *Sodiomyces* were isolated from habitats with alkaline conditions (pH of ≥10), such as soda salterns and the edges of the soda lakes of Russia, Mongolia, Kenya, and Tanzania. All species of this genus (*S. tronii*, *S. magadii*, and *S. alkalinus*) are obligate alkaliphiles [5,8,9].

The demand for novel and improved medicine from biological sources to cater to the biopharmaceutical sector has increased significantly in recent years. Investigating unexplored ecological units on the globe synergizes with the concept of investigating the least or not explored species of microbes [10]. Alkaliphilic fungi show a great potency to produce enzymes and antibiotics due to their ability to survive and persist in extreme environmental conditions [11,12,13]. From previous studies on alkaliphilic and alkali-tolerant fungi, the *Emericellopsis* genus was found to produce novel peptides with antimicrobial and antitumor activity [14,15].

The screening of metabolites of eight obligate alkaliphilic strains of *Sodiomyces alkalinus* obtained from soda soils has revealed antifungal activity against different fungal taxa, including human pathogenic isolates. As a result of the determination of the spectrum and yield of antibiotic compounds, a promising producer of *Sodiomyces alkalinus* was selected from the most active strains: 8KS17-10. This producer exhibited antifungal activity against opportunistic fungi, as well as pathogenic clinical isolates of molds and yeasts—pathogens of systemic mycoses. We have previously identified compounds as antimicrobial peptides based on the totality of the identified structural features (molecular weight, absorption ratio at certain wavelengths) [16].

In this study, we focus on the purification, identification, and antifungal activity of a novel biomolecule that has been identified as a hydrophobin and denoted as Sa-HFB1. Hydrophobins are a large family of small, cysteine-rich proteins produced by filamentous fungi. HFBs are characterized by their small size and their amphipathic nature at hydrophilic/hydrophobic interfaces, and are currently attracting great interest from the biotechnology industry [17]. This is the first time that the antifungal activity of a hydrophobin has been described.

## 2. Materials and Methods

### 2.1. Strains of Sodiomyces alkalinus

The objects of study were two alkaliphilic strains of *S. alkalinus* Grum-Grzhimaylo, Debets, and Bilanenko (Plectosphaerellaceae, Glomerellales, Hypocreomycetidae, Sordariomycetes, Pezizomycotina, Ascomycota): the F11 ex-type strain (=CBS 110278) [9] and the 8KS17-10 strain (=SLF 0117.0810). The 8KS17-10 strain was identified as *S. alkalinus* based on morphological features (Appendix A) and on a comparison of the sequences within the ITS region of strain 8KS17-10 with those of an ex-type strain (NCBI ID ON146325 and JX158405, respectively, with a 99.51% similarity score) and sequences in the GenBank database (https://www.ncbi.nlm.nih.gov/genbank accessed on 10 April 2022) [9]. The cultures were obtained from the Fungi of Extreme Conditions collection of the Department of Mycology and Algology, Faculty of Biology, Lomonosov Moscow State University (Moscow, Russia).

### 2.2. Cultivation of the Sodiomyces alkalinus and the Extraction of the Hydrophobin

Fungi were cultivated according to the previous protocol on a special alkaline medium, containing, per liter, the following: (1) Na_2_CO_3_—24 g, NaHCO_3_—6 g, NaCl—5 g, KNO_3_—1 g, and K_2_HPO_4_—1 g; and (2) malt extract—17 g, yeast extract—1 g. Components 1 and 2 were autoclaved separately for 20 min at 120 °C and mixed together after cooling, which resulted in a final pH of 10 [9]. Fungi were grown in 500 mL flasks for 14 days at 26 °C in Erlenmeyer flasks [15]. The culture liquid (CL) was separated by filtration through membrane filters MF-Millipore, Merk (Darmstadt, Germany) on a Seitz funnel under vacuum. To isolate the antibiotic substances, the CL of the producers was extracted three times with ethyl acetate in an organic solvent/CL ratio of 1:5. The extracts were evaporated under vacuum on a Rotavapor rotary evaporator (Buchi, Flawil, Switzerland) to dryness at 42 °C, the residue was dissolved in aqueous 70% ethanol, and the alcohol concentrates were obtained.

### 2.3. Purification and Identification of the Hydrophobin

#### 2.3.1. HPLC Analysis

Further separation of the active fractions (after extraction) was carried out via analytical reversed-phase high-performance liquid chromatography (RP-HPLC) with an XBridge 5 µm 130 A column with a size of 250 × 4.6 mm (Waters, Maynooth, Ireland) in a growing linear gradient of the acetonitrile concentration as a mobile phase (eluent A, 0.1% trifluoroacetic acid (TFA) (HPLC, Sigma-Aldrich, Darmstadt, Germany)) (in water MQ; eluent B, 80% acetonitrile in 0.1% aqueous TFA) at a flow rate of 950 L/min. Ultragradient acetonitrile (Panreac, Barcelona, Spain) and TFA (HPLC, Sigma-Aldrich, Darmstadt, Germany) were used for the RP-HPLC. The substances to be separated were determined at a wavelength of 214 nm in the concentration gradient of eluent B: 16–28% in 12 min; 28–55% in 27 min; 55–75% in 20 min; and 75–85% in 10 min. This was followed by isocratic elution for 25 min. The rechromatography of an active compound was performed by using a Jupiter C5 4.6 × 250 mm analytical HPLC column (Phenomenex, Torrance, CA, USA) at the same gradient conditions and at a flow rate of 1 mL/min. The absorbance (D) was determined at a wavelength of 214 nm and a mobile phase flow rate of 4 mL/min. The fractions obtained during the RP-HPLC, which correspond to individual peaks, were collected manually, and the excess of the organic solvent (acetonitrile) was then removed via evaporation in a SpeedVac vacuum centrifuge (Thermo Fisher Scientific, Waltham, MA, USA) and lyophilized (Labconco, Kansas, MO, USA) to remove residual amounts of TFA [16]. The spectrum of the antimicrobial action of the substances contained in the fractions was determined via disk diffusion, as described below.

#### 2.3.2. RP-HPLC Analysis of the Hydrophobin Sa-HFB1 Tryptic Hydrolysate

The hydrophobin obtained after the trypsinolysis reaction mixture was analyzed with the RP-HPLC method described in [18].

#### 2.3.3. Matrix-Assisted Laser Desorption Ionization Mass Spectrography (MALDI-MS) and Tandem Mass Spectography (MS|MS)

Analysis was performed on a MALDI time-of-flight (ToF) mass spectrometer (Ultraflex II Bruker, Bremen, Germany) equipped with a neodymium-doped (Nd) laser. The [M+H]^+^ molecular ions were measured in the reflector mode; the accuracy of the mass peak measurement was within 0.005%. The identification of the proteins was carried out by a peptide fingerprint search using Mascot software (Bremen, Germany) [19] through the NCBL mammalian protein database with the indicated accuracy. The search allowed for the possible oxidation of methionine by environmental oxygen and the modification of cysteine with acrylamide; where a score was >71, protein matches were considered significant (pb0.05).

#### 2.3.4. Hydrophobin Sa-HFB1 Peptides Preparation for MALDI-TOF MS Analysis

An aliquot of the peptide solution (1 μL) obtained by trypsinolysis was applied on the steal target, where it was mixed up with 0.3 μL of 2,5-dexidroxobenzoic acid (Sigma-Aldrich, Darmstadt, Germany) in a solution of 20 mg/mL 20% acetonitrile (Sigma-Aldrich, Darmstadt, Germany) in 0.5% TFA (Sigma-Aldrich, Darmstadt, Germany). Drops on the target were dried at the atmospheric temperature and pressure.

#### 2.3.5. Hydrophobin (Sa-HFB1) Derivate Preparation

A sample of the hydrophobin Sa-HFB1 in ethanol (20 μL) with a concentration of 1 μg/μL was entirely dried in a rotor evaporator (Labconco, Kansas, MO, USA). Then, 32 μL of a 100 mM NH_4_HCO_3_ water solution was added, and the mixture was heated up for 3 h. After this was performed, 4 μL of 2 M DTT (Serva, Heidelberg, Germany) was added, and it was stored at room temperature for 30 min. Finally, a portion of 0.6 M IAA (Sigma-Aldrich, St. Louis, MO, USA) in a volume of 4 μL was added, and the total solution was stored for 30 min at room temperature in full darkness.

#### 2.3.6. Hydrophobin (Sa-HFB1) Proteolysis

To the solution obtained in the derivatization reaction process described in the previous paragraph, 40 μL of modified trypsin (Promega, Madison, WI, USA) in a 100 mM NH_4_HCO_3_ solution was added at a ratio of enzyme to substrate of 1:50. Thus, the obtained mixture was stored at 37 °C for 1 h, after which some trypsin was additionally added to the reaction mixture, and then the solution was stored for 4 h at 37 °C.

#### 2.3.7. Amino Acid Analysis of a Hydrophobin Sa-HFB1 Sample

Before analysis, a probe was used to perform protein hydrolysis with the method presented in [20], and then was analyzed in agreement with the method exhibited in [21].

#### 2.3.8. Edman Sequencing

Automated N-terminal sequencing was performed on a PSSQ-33A sequencer (Shimadzu Corp., Kyoto, Japan) according to the manufacturer’s protocol. The identification of the PTH derivatives of amino acid residues was conducted was carried out using the LabSolutions software version 1.10(ROM 3.0) (Shimadzu Corp., Kyoto, Japan).

### 2.4. Sequences Analysis and Primer Design

Based on the hydrophobin Sa-HFB1 DNA sequence obtained from the *S. alkalinus* F11 whole-genome sequence (NCBI ID XM_028612330.1 and NW_021167086.1), new primers were designed to amplify the corresponding genomic regions in various strains of *Sodiomyces* species. Sequences were aligned using the Geneious program (Geneious version 7.1.5; Biomatters Ltd., Auckland, New Zealand) [22] with the default settings for multiple alignments. All of the primers used were designed by using the PRIMER3 online tool [23] and Geneious software (Geneious version 7.1.5; Biomatters Ltd., Auckland, New Zealand) Primers were designed from the intron boundaries of the exons to amplify the coding regions. The criteria for the selection of the best primer combination was a clear sequence chromatogram. The final primer pair used in the study was Hyd1F (5′-AATCCCTTATCTTTTCCAGCTTAACC-3′) and Hyd580r (5′-CAGACACGACGAGTTCGACA-3′).

### 2.5. Hydrophobin Gene Sequencing

The genomic DNA was isolated using a set of DNeasy PowerSoil Kit reagents (Qiagen Inc., Carlsbad, CA, USA), following the manufacturer’s instructions. The final volume of the 50 µL PCR mix included the following: 25 µL of 2X PCR Master Mix (Thermo Fisher Scientific, Waltham, MA, USA), 0.5 µM of each primer, and 1–100 ng of isolated DNA and water (nuclease-free). PCR was performed according to the following scheme: (1) 94 °C for 5 min, (2) 33 cycles with alternating temperature intervals: 94 °C for 1 min, 60 °C for 1 min, and 72 °C for 1 min, and (3) 72 °C for 7 min. The DNA fragments were sequenced by the Sanger method on an Applied Biosystems 3500 Series Genetic Analyzer. Sequence data were submitted to GenBank (NCBI ID ON149453).

### 2.6. Biological Assays

#### Antifungal Activity

The antifungal activity was measured by the disk diffusion method. Disks of 6 mm diameter, containing 40 µL of sample, were deposited on PDA agar plates (Sigma-Aldrich, St. Louis, MO, USA). The spectrum of the antifungal activity of the CL, extracts, and individual compounds was determined on test cultures of mycelial and yeast fungi from the collection of cultures of the Gause Institute of New Antibiotics (Moscow, Russia). Opportunistic mold and yeast test cultures of the fungal species *Aspergillus fumigatus* KPB F-37, *Penicillium brevicompactum* VKM F-4481, *A. niger* INA 00760, and *Candida albicans* ATCC 2091 were used. The diameters of the inhibition zones were measured after 24 h at 28 °C. The sensitivity of the test organism was controlled with standard disks containing amphotericin B (AmpB), itraconazole (IZ), fluconazole (FZ), and voriconazole (VOR) (40 μg/disk) as positive controls.

The spectrum of antifungal action was also evaluated on clinical isolates of molds and yeasts pathogens of opportunistic pneumomycosis of the bronchi and lungs—in tuberculosis patients with multi-resistance to the antibiotics–azoles used in clinical practice, from the collection of the mycological laboratory of the Moscow City Scientific and Practical Center for Tuberculosis Control (Russia): *Candida albicans* 1582 m, *Pichia kudriavzevii* (former *C. glabrata*) 1402 m, *Nakaseomyces glabrataa* 1447 m, *C. parapsilopsis* 571 m, *C. tropicalis* 156 m, *Cryptococcus neoformans* 297 m, *Aspergillus fumigatus* 390 m, and *A. niger* 219 m.

The minimal inhibitory concentration (MIC) value of each individual compound was determined using the broth twofold microdilution method according to the CLSI/NCCLS documents M27-A3 and M38-A2. The MIC was determined for the yeast fungi *Candida albicans* ATCC 2091, *C. albicans* 1582 m, and *Cryptococcus neoformans* 297 m, as well as for 48 h for the molds *Aspergillus niger* INA 00760, *A. fumigatus* KPB F-37, and *A. fumigatus* 390 m. The minimum inhibitory concentrations (MIC) were determined by a two-fold serial microdilution method in a liquid culture medium RPMI 1640 with Lglutamine without sodium bicarbonate for fungi in accordance with the requirements of the Institute of Clinical and Laboratory Standards [24,25].

The experiments were carried out in three to five replicates. Statistical processing of the results and the assessment of the reliability of the differences in mean values were carried out according to the Student’s *t*-test for a probability level of at least 95% with Microsoft Excel 2010 and Statistica 10.0 software (Palo Alto, CA, USA).

## 3. Results

### 3.1. Isolation of the Active Polypeptide from Culture Broth Extract

According to previous data, a high level of active compounds from culture liquids was obtained for the *S. alkalinus* 8KS17-10 strain. Analytical separation by reversed-phase HPLC for the ethyl acetate extract was carried out according to the data described earlier [16]. The maximum amount of an individual peak was determined for the 8KS17-10 strain, and it was selected for further investigations. It was shown that an active compound can be attributed to the group of peptides based on the totality of the identified structural features (molecular weight, absorption ratio at certain wavelengths). Previously, a component with a retention time of 31.171 min with strong antifungal activity against a number of clinical fungal and yeast isolates (*Candida* spp., *Aspergillus* spp., and *Cryptococcus neoformans*) was shown [16].

The peptide production yield from the culture liquid and mycelium extracts of *S. alkalinus* on the alkaline medium was achieved in a period of fermentation in 14 days of about 25.54 ± 1.4 mg/L. To obtain the compound for further structural analyses we made three single-type chromatographic separations, and the target peak was collected manually (Figure 1). In the following steps, this compound was rechromatographed to confirm its purity, and an individual molecule was prepared for investigation by analytical methods.

### 3.2. Structural Analysis

Previously, it has been suggested that the molecule studied is typical of a polypeptide; therefore, its initial structure analysis was conducted by automated Edman sequencing. After ten cycles, a predominant N-terminal amino acid sequence for the analyzed peptide was composed, ^1^TYIAXPISLY^10^ (Appendix A). Searching for potential homologies among NCBI databases using the BLASTP algorithm led to complete matching with the fungal hydrophobin F11 (NCBI ID ROT36721.1/XP_028464527.1). These data were obtained using previously annotated whole-genome shotgun sequences for the ex-type strain of *S. alkalinus*, F11 [26].

The complete matchup for the N-terminal sequence of the isolated peptide with the *S. alkalinus* fungal hydrophobin F11 allowed us to specify that the fifth amino acid residue is cysteine, which can be visualized as a PTH-s-beta-4-pyridylethylated derivative [27,28]. The isolated polypeptide is found and denominated as Sa-HFB1. Hence, the complete amino acid sequence for the precursor of *S. alkalinus*, the class II hydrophobin F11 (NCBI ID XP_028464527.1), is ^1^MKFIAVVAALTASLAMAAPTESSTDTTYIA**CPISLYGNAQCCATDILGLANLDCESPTDVPRDAGHFQRTCADVGKRARCCAIPVLGQALLCIQP**AGAN^99^ [26]. Based on NCBI annotation data for ID XP_028464527.1, the current protein precursor has a calculated molecular mass of 10181.6 Da; residues 1–30 are represented a signal peptide, whereas residues 31–95 form a mature hydrophobin sequence (marked in bold), in which the cleaved primary peptide bond is located between Tre26 and Tre27. This allowed for the identification of one new hydrophobin of class II, denoted as Sa-HFB1. Thus, Sa-HFB1 typically has up to five additional amino acid residues at the N-terminus. For the identification of any possible homologies for the complete hydrophobin sequence in genome NCBI ID PRJNA196044, we used a local blast. As a result of the search, only one gene was found to correspond to the hydrophobin (located in the genomic scaffold SODALscaffold8). A further search in the transcriptional data confirmed this fact (according to scaffold SODALDRAFT_334916, NCBI ID XM_028612330.1).

As a result of the full-genome sequencing of the *S. alkalinus* F11 strain, the nucleotide sequence that encodes the hydrophobin was recently discovered and translated into an amino acid sequence in silico, since a bioinformatics assay was carried out.

Some general information calculated on the template of the predicted amino acid sequence of the hydrophobin, its molecular weight and amino acid composition, is presented in short on Figure 2.

To confirm the sequence of the hydrophobin and to examine the conservation of this protein, a sample of the hydrophobin isolate obtained by EtOAc, produced by the *S. alkalinus* 8KS17-10 strain and subsequent chromatographic purification, was analyzed.

### 3.3. MALDI-TOF MS

Individual fractions were analyzed with the MALDI-TOF MS method to obtain information about masses, consisting of a matrix. The result of the mass spectrometry is depicted in Figure 3a. Several peaks on the specter were received, but only the major one with a mass of 7588 Da ([M+H]^+^) was of interest. One of the rest peaks with a mass of 3794 Da is a binary ionized form of the major peak ([M+H]^++^), and the others were considered to be any homologues of the hydrophobin.

One real problem after the analyses concerned the mass difference between the annotated hydrophobin with a mass of 7609 Da (with the formation of disulfide bonds—7601 Da) and 7588 Da, presented at the specter. To dissolve the problem, an idea about one amino acid change in the protein was proposed, taking into account 14 Da as the difference between the major peak mass and the annotated sequence mass with a full set of disulfide bonds.

For an analysis, a table with the molecular mass differences between different natural amino acids was created, and the goal mass of 14 Da was discovered and colored in red in Figure 3b. It provided us with an opportunity to propose a set of appropriate amino acid changes, presented in Figure 3c.

### 3.4. Amino Acid Analysis

As a method for amino acid quantification in an analytical solution, an amino acid analysis was performed. Chromatograms of a standard amino acid solution and of the hydrophobin, obtained via acid hydrolysis, are presented in Figure 4a,b, respectively. In a table in Figure 4c, the amount of amino acids and the numbers of residues in the announced sequence of the hydrophobin (real values), as well as the number of residues as evaluated from the result of the amino acid analysis, assuming that only one protein exists (theoretical values), are shown. In Figure 4d, they are shown graphically as well.

The assumption that there was only one protein in a probe was generated to convert the quantity characteristics of amino acids into a number form for the next correlation analysis between the theoretical and real numbers of the corresponding amino acids. In addition, it must be reported that some amino acids degrade; still, the hydrolyses take place, and they are not presented in the results of the amino acids’ analysis. This study shows the same pattern of amino acid distribution in the theoretical and real sequences as well as the low level of dispersion in numbers of residues (Figure 4d). Taking all of this information into account, it appears that all of the proteins comprised in a probe are of the same nature and are consistent with the annotated sequences.

In addition, every mismatch in the numbers of the theoretical and real amino acid residues in addition to all of the possible amino acid changes were pointed out (Figure 5a). To provide any quantity characteristic controls, the balance conservation of the full amino acid number in a protein, the score function *d*, was proposed:$$d=\sqrt{{\left(N_{1}^{theoretical}-N_{1}^{real}\right)}^{2}+{\left(N_{2}^{theoretical}-N_{2}^{real}\right)}^{2}}$$

*N*_1_ is the number of any amino acid, which was changed to another one with the number *N*_2_. The “*theoretical*” and “*real*” indexes characterize the ways of calculating amino acid numbers. The first represents calculations based on amino acid analysis results, while the second is based on annotated sequences of hydrophobins.

This procedure was performed to reduce the number of proposed amino acid changes.

For all of the changes, presented in a table (Figure 5a), the meaning of the d function was calculated, and these changes were marked as dots on the plot (Figure 5b). The meaning of the d function, related to the normal balance of one amino acid change in the protein, was marked as a yellow dotted straight line on the plot (Figure 5b). Thus, all of the dots of amino acid changes, which are lying on the normal line (Figure 5b), are likely to be related to the examined protein.

### 3.5. Proteolysis and Amino Acid Sequencing

To study the hydrophobin in detail and to confirm the amino acid sequence of this protein, a procedure of sequencing with MALDI-TOF MS|MS was performed.

Trypsin was chosen as the enzyme for proteolysis due to the existence of several trypsinolisys sites in the protein sequence. Solving the problem of sites’ access to protease, cysteine residues were reduced with DTT and were modified with IAA (Figure 6a). Furthermore, IAA was able to modify the side chain amino group of lysine and the N-terminus of the polypeptide chain, as well the cysteine residues, and not only in a 1:1 proportion (Figure 6b). Therefore, this information was taken in account.

The probe, after modification, was analyzed with the RP-HPLC method, and all of the marked peaks (Figure 6c) were collected to perform MALDI-TOF MS and MS|MS analyses.

Modifying the radical of IAA is presented in Figure 7a, with some physicochemical information and the character of subsequent fragmentation in the MS|MS method (Figure 7b).

All of the collected fractions were analyzed with the MALDI-TOF MS method, and the corresponding specters are represented in Figure 7c (1, 2, and 3). The first specter includes one set of peaks, and the second one includes two sets. The third specter includes one set, but the mass (*m*/*z*) values indicate an unmodified hydrophobin. Inside every set, the difference in the mass between the closest peaks seems to be modifying the difference in IAA (57 Da).

All of the main peaks from the sets (1, 2, and 3) were subsequently analyzed by performing the MS|MS method, and the corresponding specters are presented in Figure 8a–c. The main peaks were chosen as the results of an annotated sequence analysis with Mascot service free web software for MALDI-TOF ion prediction [19]. They were peaks with masses of 4013 Da, 829 Da, and 2125 Da (including the masses of modifiers), and the result of the Mascot analysis is presented in Figure 8d. All of the theoretical ions (Figure 8d) were found in the specters (Figure 8a–c).

Every specter of the MS|MS analysis of the corresponding masses, 829, 2125, and 4013 Da, was sequenced, using tools of Bruker Daltonics flexAnalysis 3.4 software (Bruker, Billerica, MA, USA). The first two masses were aligned to the corresponding fragment (marked on the MS|MS specters) of the annotated sequence, but the mass of 4013 also seems to be the fragment with one amino acid change in the 29 position (Figure 8c).

Thus, an amino acid change was observed in the entire sequence fragment for the *S. alkalinus* 8KS17-10 strain, different from the one for the full-genome sequencing of the ex-type strain of *S. alkalinus*, F11. The registered change is E29D, and this result correlates with the results of the MALDI-TOF MS analysis of the probe in addition to the amino acid analysis.

### 3.6. Sequence Analysis of the S. alkalinus Hydrophobin

The genome of *S. alkalinus* (NCBI ID GCA_003711515) contains one gene encoding for a hydrophobin. No already-characterized hydrophobins of different Ascomycetes species show significant similarities in their protein and nucleotide structures. The gene has a distinct structure, containing three introns and two exons. The corresponding DNA sequence information was translated into polypeptide sequences and showed a predicted protein size of 99 residues.

A nucleotide analysis of the hydrophobin gene, using designed Hyd1F/Hyd580r primers of the F11 strain, show 100% similarity with the known hydrophobin sequence from the annotated genome of the *S. alkalinus* F11 ex-type strain (Figure 9).

The *S. alkalinus* 8KS17-10 strain sequences of the hydrophobin gene (NCBI ID ON149453) contained two substitutions: The substitution of C to A at the 369 position was in an intron, although it did not alter the amino acid sequence. The other, a G to C substitution at codon 165, was in an exon, resulting in a glutamic acid (Glu or E) to aspartic acid (Asp or D) replacement (Figure 10).

### 3.7. Antifungal Activity

Previously, we have tested the antimicrobial activity of eight *S. alkalinus* strains against 10 strains of Gram-positive and Gram-negative bacteria, in addition to fungi, including either sensitive or drug-resistant isolates from ATCC, as well as resistant clinical isolates. All of the *S. alkalinus* strains demonstrated high activity against yeast and mold fungi, including collection cultures and multidrug-resistant clinical isolates [16]. The highest amount of Sa-HFB1 and antifungal activity was detected for the *S. alkalinus* 8KS17-10 strain (Appendix A).

The antifungal activity of Sa-HFB1 against opportunistic fungi and multidrug-resistant clinical isolates is shown in Table 1. The clinical pathogenic isolates *Cryptococcus neoformans*, *Nakaseomyces glabrataa*, and *C. tropicalis* were selected based on their resistance phenotypes, including their resistance to caspofungin, micafungin, fluconazole, and flucytosine. The antifungals AmpB, FZ, and VOR were used as reference drugs. Sa-HFB1 inhibits the whole panel of opportunistic and clinical isolates with the broth twofold microdilution method. The activity of Sa-HFB1 against *Aspergilllus* spp. was comparable to the reference polyene cyclic drug AmB. The inhibition zones for all of the clinical *Candida* isolates were found to be 12–14 mm for Sa-HFB1, while *Nakaseomyces glabrataa* 1447 m and *C. tropicalis* 156 m were inactive to AmpB, FZ, and VOR, respectively.

The MICs of Sa-HFB1 against opportunistic and clinical fungi ranged from 1 to 8 µg/mL and confirmed higher activity against both opportunistic and clinical isolates than has previously been reported (Table 2). The highest level of antifungal activity (MIC 1 µg/mL) was demonstrated for *Cryptococcus neoformans* 297 m.

## 4. Discussion

The population around the globe is affected every year by mycoses caused by pathogenic fungi, which can be classified as subcutaneous, cutaneous, and systemic. Clinical pathogens, such as *Candida*, *Cryptococcus*, and *Aspergillus*, lead to severe fungal infections and are a common cause of nosocomial infections, reaching mortality rates of up to 40% [29,30]. Due to the notorious increase in the incidence of IFIs globally, which result in up to 1.7 million deaths annually, numerous strategies have been employed to optimize the treatment of these deadly infections. The main reason for limited mycosis treatment options is the development of antifungal resistance or multidrug resistance (MDR), which can abolish treatment options. There are different factors leading to drug resistance and therapeutic failure: Firstly, patients with immunosuppression are more likely to fail at responding to antifungal therapy due to the lack of assistance of a robust immune response against an infection. Secondly, long-term or repeated drug exposure can also lead to the emergence of resistance. Furthermore, being exposed to agricultural fungicides with identical molecular targets to those of systemic antifungal drugs can increase the development of resistant organisms. All drug classes can develop antifungal resistance, and some fungal species can even show resistant activity against all antifungal classes. At present, researchers direct their attention to the examination of plant and microbial isolates, secondary metabolites, and newly synthesized molecules.

Hydrophobins (HFBs) are small proteins found only in filamentous fungi (Dikarya) [31,32,33,34,35,36,37]. HFBs are characterized by their small size and amphipathic nature at hydrophilic/hydrophobic interfaces, and are currently attracting great interest from the biotechnology industry [38,39,40,41]. A central attraction is the ability of a hydrophobin monolayer to reverse the nature of a surface from hydrophobic to hydrophilic, and vice versa [31,37,42]. These layers cover fungal bodies and spores in water-repelling coats [37] and influence spore dispersal, stress resistance, development, and biotic interactions [43,44,45]. In pathogenic fungi, such as *Aspergillus fumigatus* [46,47], *Metarhizium brunneum*, and *M. acridum* [43,44], HFBs are considered to be virulence factors because they reduce the exposure of pathogen-associated molecular patterns (PAMPs) and antigens to receptors of the immune system [43,44,45,48]. HFBs are also involved in symbiotic interactions, such as those between lichens and mycorrhizae [17,49,50,51]. HFBs play a role in development and morphogenesis in the majority of filamentous fungi and influence spore properties [33,34,52]. To reveal HFBs that are associated with the sporulation of *T.*
*harzianum* CBS 226.95 and *T. guizhouense* NJAU 4742, the expression of respective genes during the three stages of fungal development has been tested. The results showed that two genes were highly expressed during the formation of aerial mycelium and remained highly active during conidiation [53,54,55,56].

Among biocontrol fungi, some studies have revealed that the participation of HFBs in biocontrol processes could contribute to plant growth and elicit plant defense reactions. *T. asperellum* mutants that lack the TasHyd1 hydrophobin gene are severely impaired in terms of root attachment and colonization, and these phenotypes are recovered by the complementation of TasHyd1, indicating that this protein contributes to the interactions between *Trichoderma* and its host plant [55,57,58,59]. In *Clonostachys rosea*, the Hyd3 class II hydrophobin can influence root colonization ability [45]. Similarly, the HYTLO1 class II hydrophobin in *T. longibrachiatum* can directly inhibit both the spore germination and hyphal elongation of *Botrytis cinerea* and *Alternaria alternata* in vitro, as well as enhance tomato plantlet development. HYTLO1, which has multiple roles in and effects on treated plants, was able to trigger a nicotinic acid adenine dinucleotide phosphate-mediated Ca^2+^ signaling pathway in *Lotus japonicus*, highlighting a possible mechanism underlying its action. Zhang et al. demonstrated the biocontrol functions of HFB2-6, a class II hydrophobin of *T. asperellum* ACCC30536 [53,56]. The HFB2-6 hydrophobin gene was expressed in *Escherichia*
*coli*, and the rHFB2-6 recombinant protein was found to affect the transcription of poplar defense-related genes. HFBII-4 from *T. asperellum* can enhance the resistance of *Populus davidiana* × *P. alba* var. *pyramidalis* to *A. alternata* phytopathogenic fungi. In summary, the class II hydrophobin gene HFBII-4 upregulated the expression of growth-related, disease resistance, and defense response genes in PdPap poplars [44,56].

Fungi growing at extreme pH values are of scientific interest for the general study of fungal adaptive evolution, as well as for the evaluation of their potential in producing commercially valuable substances. Obviously, fungi adapting to alkalinity must have metabolic pathways that have become modified with respect to those seen in related neutrophilic fungi [1,2,3,5,9,60]. Alkaliphilic *Sodiomyces* species are fundamentally different from alkali-tolerant ones in terms of their mechanisms of adaptation. They accumulate trehalose in the cytosol and phosphatidic acids in the membrane lipids, whereas alkali-tolerant fungi contain these compounds in low amounts. In addition, adaptations to alkaline environments are required for structures involved in exporting metabolites such as antibiotics, for domains of membrane transporters exposed to ambient environments, and for the regulation of gene expression by an ambient pH. The life cycle of *S. alkalinus* and some possible mechanisms of its adaptation to a high pH combined with salinization have been investigated recently at the cytomorphological level [61], while the biochemical mechanisms have not been studied sufficiently yet [62,63,64]. Our experiments to evaluate the antifungal activity and identification of an active hydrophobin might offer a clue as to the possible ecological role and biological activity of alkaliphilic *S. alkalinus* in alkaline soils.

The structural, MALDI MS|MS, and NCBI annotation data analyses helped to identify Sa-HFB1 as a class II hydrophobin. Hydrophobin genes may have SNPs, resulting in different amino acid sequences in synthesized HFBs of different strains. The Sa-HFB1 production yield of about 25.54 ± 1.4 mg/L from the culture liquid of *S. alkalinus* on an alkaline medium was achieved in a period of fermentation of 14 days. This value has been found to be typical of class II HFBs secreted by wild-type fungi into an extracellular medium [40,41,51,58,65]. Previous research has reported the amount of HFBII of wild-type *T. reesei* to be around 30 mg/L, and around 200 mg/L for genetically engineered *T. reesei* [38]. This is the first time that *S. alkalinus* was explored for the isolation of HFBs.

In our previous studies, we demonstrated the antifungal activity of different species of conditionally pathogenic molds of *Aspergillus* spp. for the *S. alkalinus* 8KS17-10 strain. We have previously identified the compound as an antimicrobial peptide based on the totality of the identified structural features (molecular weight, absorption ratio at certain wavelengths). In the present study, we also showed that the active compound is an antimicrobial peptide and highly effective towards pathogenic molds. Due to the strong antifungal property exhibited by Sa-HFB1, it has been evaluated to further antifungal activity against pathogenic clinical fungi and yeasts. To the best of our knowledge, this is the first report of a direct antifungal effect of class II HFBs from alkaliphilic fungi against non- and clinical pathogen’s isolates.

The hydrophobin, produced by the *S. alkalinus* 8KS17-10 strain, was isolated and purified chromatographically, analyzed, and sequenced using a trypsinolisys assay in addition to the MALDI-TOF MS and MS|MS methods. As a result, we report the convergence of amino acids between the experimental and annotated sequences, obtained by the genome sequencing of the *S. alkalinus* F11 strain, to within one amino acid in the 29 position. Thus, we declare more concretely the change of glutamic acid to aspartic acid in the 29 position of the protein (E29D), as well as the fact that this region has a low level of residue conservation, so amino acids can differ there from strain to strain.

## 5. Conclusions

There are *Sodiomyces* fungi that can secrete active molecules, such as HFBs. Therefore, discovering novel FBs from unstudied alkaliphilic fungi and understanding the possible role of these molecules is essential to realize the full biotechnological potential of these proteins. In this study, we isolated and identified, for the first time, a class II hydrophobin from *S. alkalinus*, denominated as Sa-HFB1. Moreover, this study is the first to report its antifungal activity towards opportunistic and pathogenic fungi.

## Figures and Tables

**Figure 1 jof-08-00659-f001:**
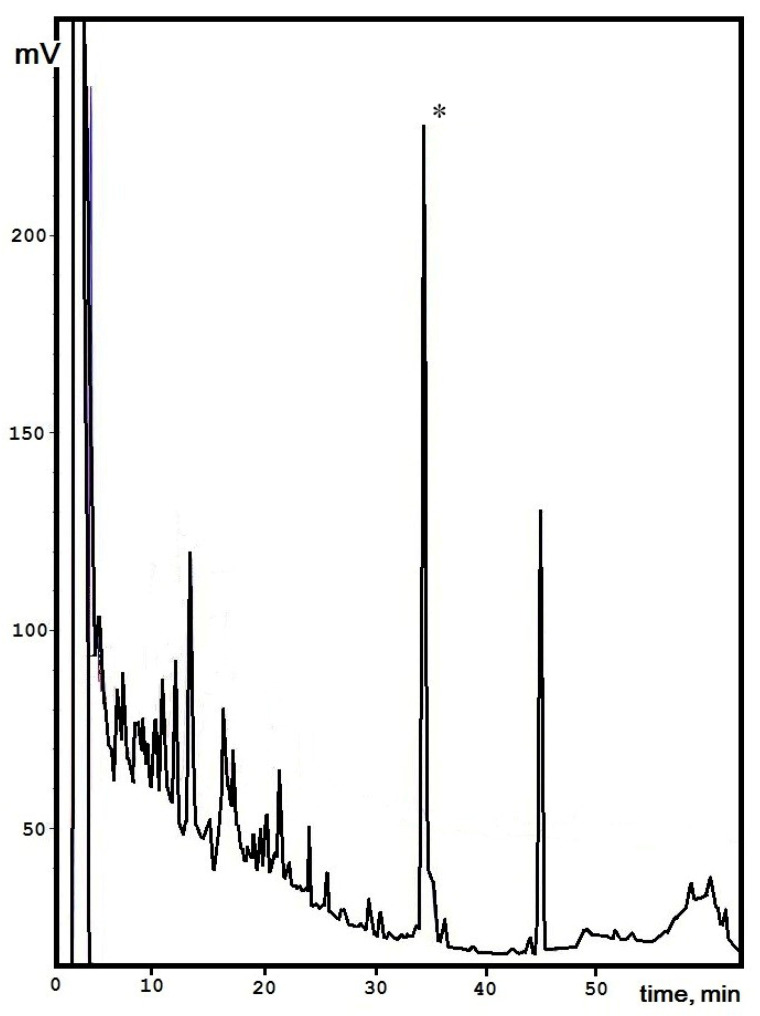
Isolation of Sa-HFB1 from *S. alkalinus* ethyl acetate concentrate. The target peak is indicated by in asterisk.

**Figure 2 jof-08-00659-f002:**
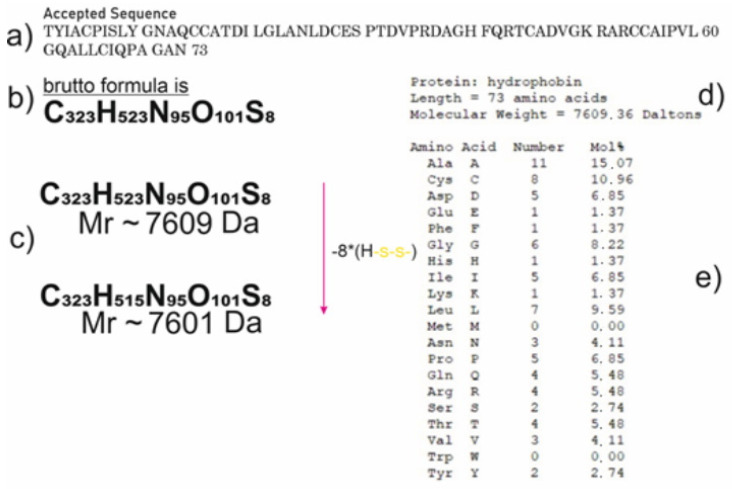
Physicochemical properties of the hydrophobin. (**a**) The sequence of the hydrophobin, annotated under the result of the full-genome sequencing of the ex-type strain of *Sodiomyces alkalinus*, F11. (**b**) Atom composition of the goal protein, impressed in numbers of corresponding atoms. (**c**) Chemical transition in mass and atom composition due to all of the disulfide bonds forming. (**d**) Information about the exact mass of the protein. (**e**) The amino acid composition of the hydrophobin.

**Figure 3 jof-08-00659-f003:**
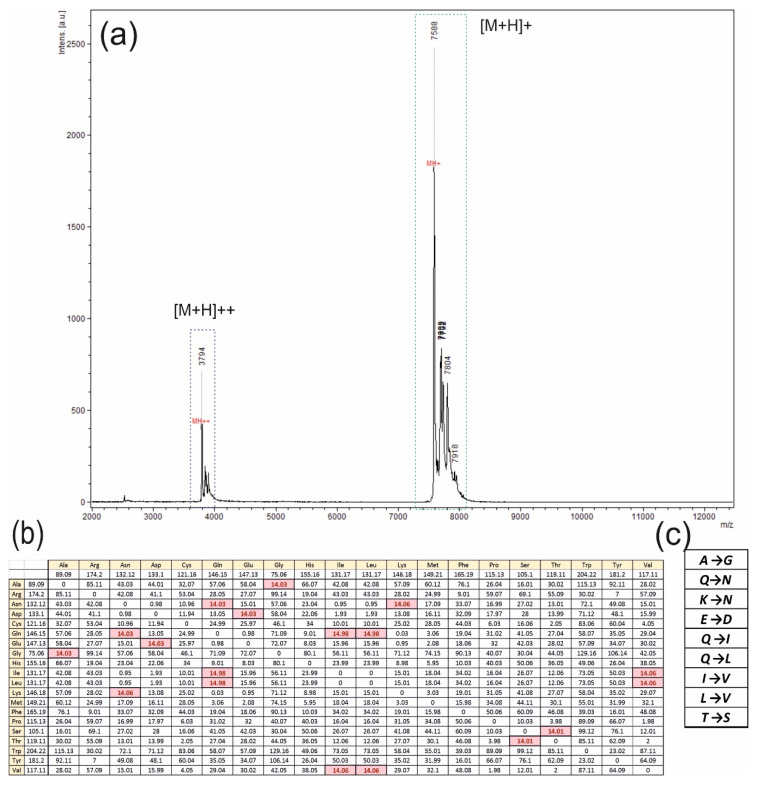
Analysis of peaks with the MALDI-TOF MS method. (**a**) The spectrum with two fields of peaks is presented. The main peak, with a mass of 7588 Da, is highlighted with others in one set in the dotted rectangle colored in green, and another set of peaks with the main peak mass of 3794 Da is highlighted with the blue dotted rectangle. Symbols [M+H]^+^ and [M+H]^++^ mean the masses of the molecular ions plus proton, the single-charged and double-charged ions, respectively. (**b**) The table, including information about the molecular mass differences between each of the natural amino acids; results are presented in Da units, and the goal of 14 Da, meaning between corresponding amino acids, is colored in red. (**c**) The set of possible amino acid single changes in the hydrophobin, calculated over the molecular masses, is presented.

**Figure 4 jof-08-00659-f004:**
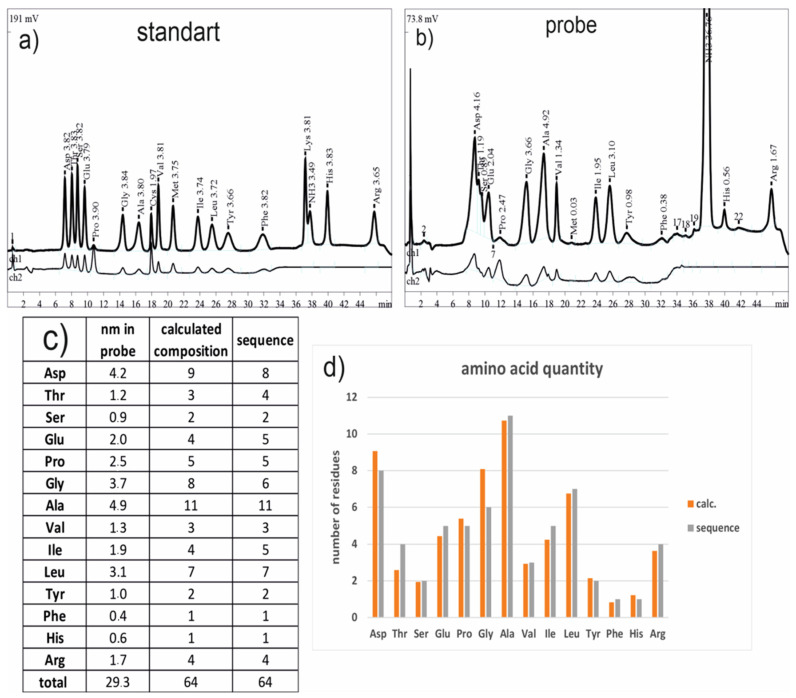
The results of the amino acids’ analysis. (**a**) Chromatogram of the standard mixture of amino acids examined at two wavelengths: 440 nm (ch1) and 590 nm (ch2). (**b**) Chromatogram of the real amino acid mixture, obtained via the hydrolysis of the hydrophobin with mineral acids (ch1 and ch2 are the same as those in (**a**)). (**c**) A table with the amino acids’ quantification and transformation into numbers of amino acids in a hypothetical protein in a sample (calculated composition), and the amino acid numbers in a query annotated sequence (sequence). Some of the amino acids are not presented in the table (Lys, Met, Trp, and Cys) due to their degradation; still, the mineral acid hydrolysis take place, and the others (Gln and Asn), at the same conditions, transform into Glu and Asp, correspondingly. (**d**) Information about the amino acid distribution in the protein examined via amino acid analysis (orange bars) and the residue numbers in the annotated sequence (grey) is graphically presented.

**Figure 5 jof-08-00659-f005:**
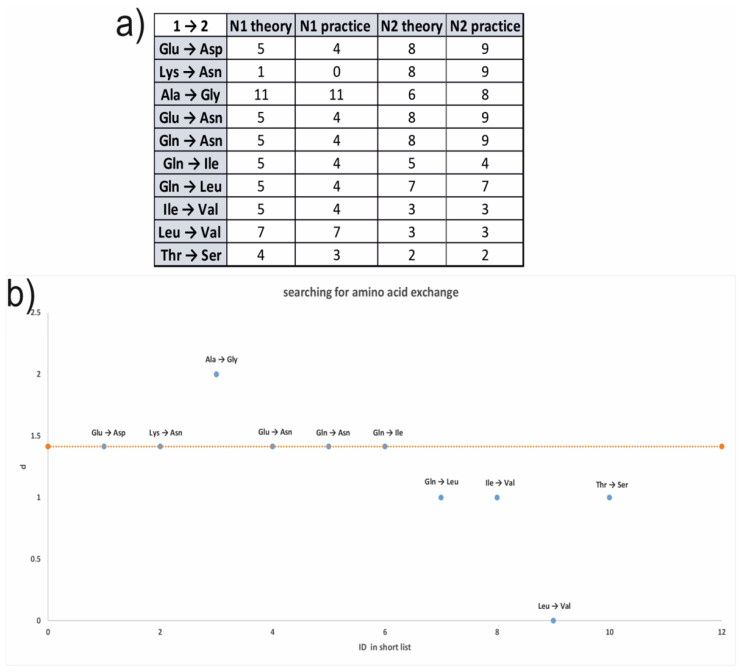
Amino acid change research over the amino acid analysis results. (**a**) A table including all of the possible amino acid changes, where the initial amino acid, “1” (presented in the annotated sequence), changed to another one, “2” (presented in the results of research). *N*_1_ and *N*_2_ are the numbers of corresponding amino acids. (**b**) The *d* function plot is depicted. Each amino acid change is displayed on the plot with the calculated meaning of the *d* function. The *d* function value, related to the normal proportion of a one-letter change in the protein, is marked on the plot as a yellow dotted straight line.

**Figure 6 jof-08-00659-f006:**
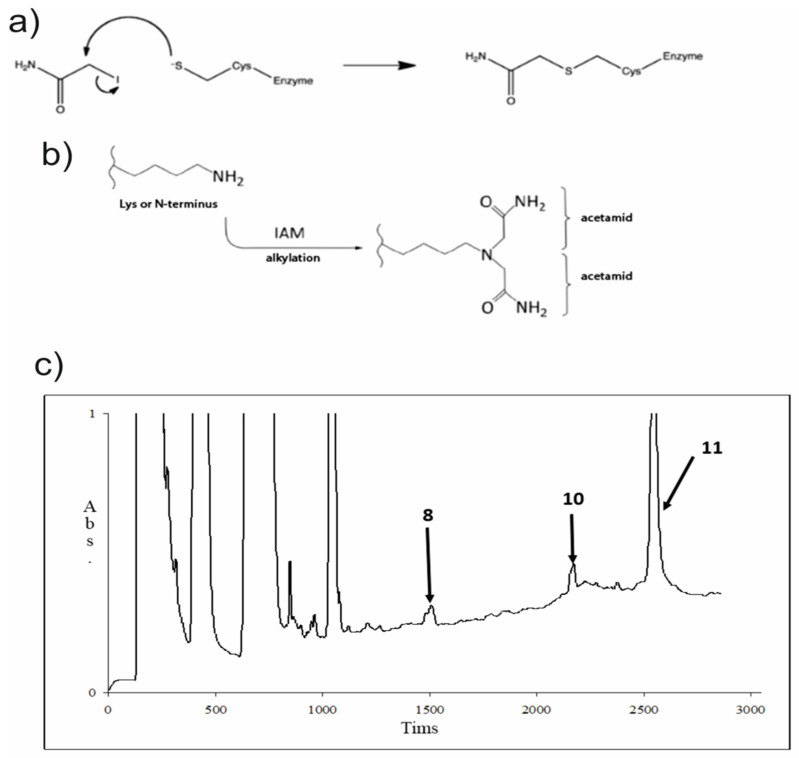
Protein modification and trypsinolisys results. (**a**) The chemical reaction of cysteine modification with IAA is presented. (**b**) The adverse reactions of the lysine side chain amino group and the N-terminus of the polypeptide modification with IAA are depicted. (**c**) An RP-HPLC chromatogram of the sample, performed after the trypsinolisys took place. The collected peaks are marked with corresponding numbers.

**Figure 7 jof-08-00659-f007:**
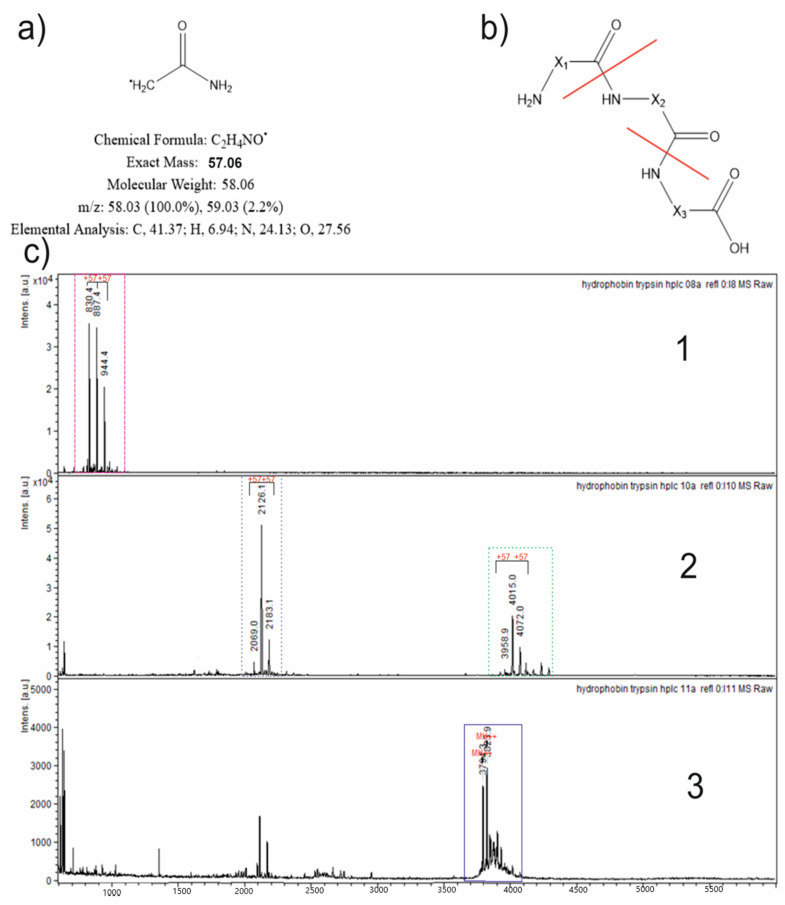
MALDI-TOF MS analysis of the collected fractions. (**a**) An acetamide radical as a modifying unit is presented with some physicochemical information. (**b**) The character of probable fragmentation in the MS|MS specters and the broken bonds are marked with red lines. (**c**) The mass specters of chromatographically collected fractions are presented. Peaks within every set that differ between each other by 57 Da (modifying unit) are marked with black lines and indexes “+57” above the peaks. For specters 1 and 2, the peak sets were highlighted in dotted rectangles colored in corresponding colors. The set in specter 3 is highlighted in a continuous rectangle as the mass of the unmodified hydrophobin.

**Figure 8 jof-08-00659-f008:**
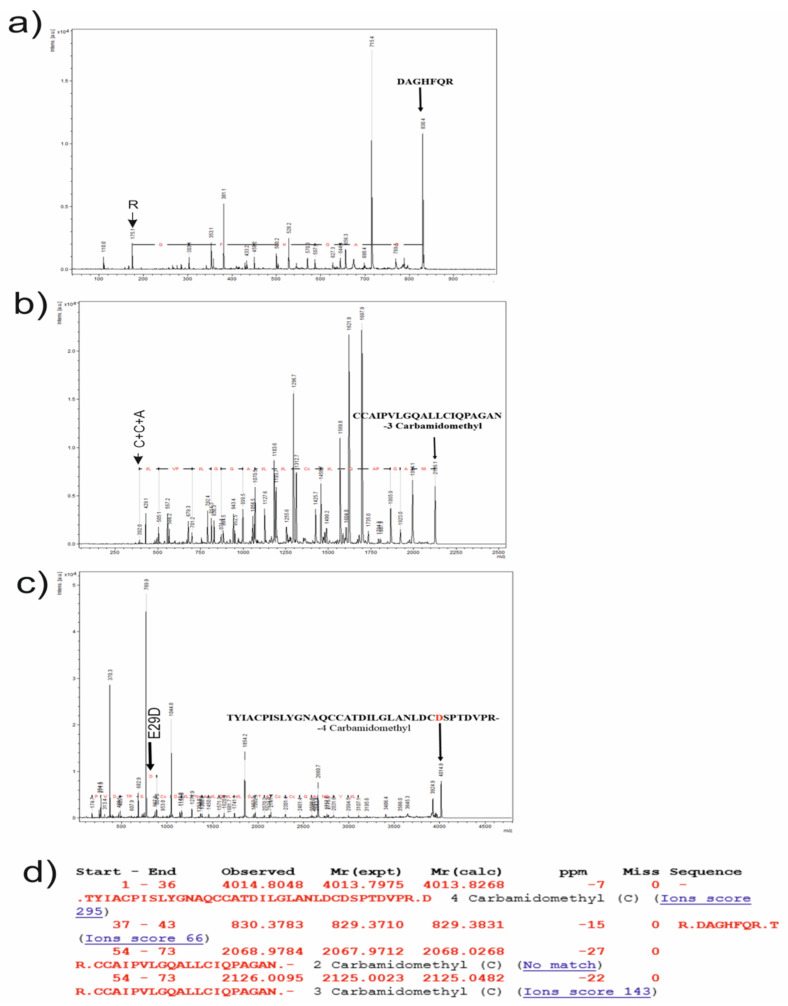
Sequencing of the hydrophobin. (**a**–**c**) MALDI-TOF MS|MS specters of different fragments of the hydrophobin, obtained with the utilization of trypsinolisys and separated with the RP-HPLC method, are presented. The characters of fragments are pointed out in black above the rightmost peak in every specter. In addition, the annotation to other peaks is exhibited with black labels above the corresponding peaks, and the amino acid change is indicated with a label and arrow. The process of sequencing is exhibited with a black line joining together the corresponding peaks with a red annotation in amino acid (acids) symbol(s). (**d**) The probable fragments of trypsinolisys with different types of modification, which were predicted with Mascot service free web software for MALDI-TOF ion prediction correlated with the masses exhibited on the specters above, are presented.

**Figure 9 jof-08-00659-f009:**
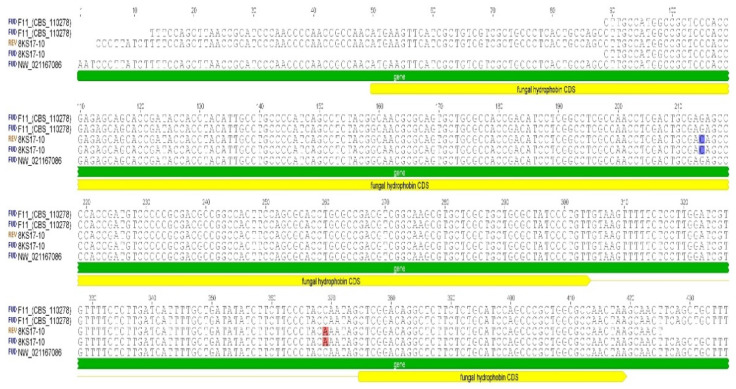
Alignment of hydrophobin DNA sequences. The alignment shows single-nucleotide polymorphisms (SNPs) in the *S. alkalinus* 8KS17-10 strain. An exon substitution is marked by blue highlighting, and an intron SNP by red highlighting.

**Figure 10 jof-08-00659-f010:**
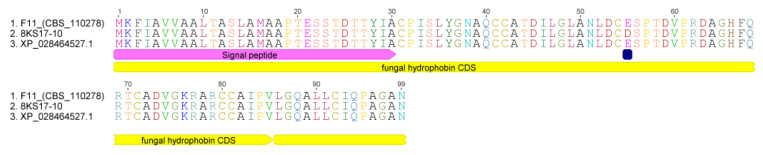
Alignment of the hydrophobin’s amino acid sequences. The alignment shows an E to D amino acid replacement at position 29 in the mature hydrophobin sequence (amino acids 1–30 are signal peptides).

**Table 1 jof-08-00659-t001:** The activity of purified Sa-HFB1 on the growth of opportunistic and clinical pathogenic fungi isolates measured by a disk diffusion assay.

Zone, mm				
Compound, 40 µg/disk				
Strain	Sa-HFB1	AmpB	FZ	VOR
*Candida albicans* 1582 m	15 ± 0.1	10 ± 0.6	0	10 ± 0.6
*Pichia kudriavzevii* 1402 m	16 ± 0.3	15 ± 0.1	0	10 ± 0.6
*Nakaseomyces glabrataa* 1447 m	12.5 ± 0.2	0	0	0
*C. tropicalis* 156 m	14 ± 0.1	0	0	0
*C. parapsilosis* 571 m	14 ± 0.2	18 ± 0.3	0	0
*Cryptococcus neoformans* 297 m	30 ± 0.1	18 ± 0.6	0	0
*Aspergillus fumigatus* 390 m	12 ± 0.5	9 ± 0.6	0	0
*A. niger* 219	14 ± 0.1	15 ± 0.8	0	11 ± 0.6
*Penicillium brevicompactum* VKM F-4481	15 ± 0.5	17 ± 0.4	0	10 ± 0.9

**Table 2 jof-08-00659-t002:** Minimal inhibitory concentration of Sa-HFB1 compared with target antifungal drugs.

Minimal Inhibitory Concentration (MIC, µg/mL)				
Compound				
Strain	Sa-HFB1	AmpB	FZ	IZ
*Aspergillus niger* 219	8	1	R	4
*A. fumigatus* 390 m	8	1	R	0.5
*Candida albicans* ATCC 2091	4	1	>64	4
*C. albicans* 1582 m	4	2	>64	4
*Cryptococcus neoformans* 297 m	1	0.5	16	0.5

R—resistant; AmpB—amphotericin B; FZ—fluconazole; and IZ—itraconazole.

## Data Availability

All sequence data are available in NCBI GenBank following the accession numbers in the manuscript.

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
