# Peer review of "Isolation and Characterization of a Novel Hydrophobin, Sa-HFB1, with Antifungal Activity from an Alkaliphilic Fungus, Sodiomyces alkalinus"

_jof, 2022, doi:10.3390/jof8070659_

Round 1
Reviewer 1 Report
This study involved purification and characterization of the hydrophobin isolated from the alkaliphilic fungus Sodionyces alkalinus. The result of study showed novelty and very interesting to the medical area. There are some concerning issues as followings.
Major concerns
- The antifungal susceptibility testing should be done again. The results in table 2 is not completed. C. neoformans297m had not tested with the conventional drugs so there was no control for this organism. I also concerned about inconsistent results between disc diffusion and broth dilution method for C.albicans 1585m. The disc diffusion method showed that Sa-HFB1 has larger zone, but it showed equivalent MIC. Would it be possible if the discrepancy came from improper techniques in the disc diffusion assay. As I saw in figure S2, the tested organisms were not spread entirely on the surface of the medium. This low inoculum size could affect the killing ability of diffused drugs. Please consider re-performing this experiment.
- Several names of the fungi included in this study need to be changes. According to a paper titled “Name changes for fungi of medical important 2018 to 2019. J Clin Microbiol 2021. 59(2): e01811-20. The name of Candida krusei and C. glabrata were changed to Pichia kudriavzevii and Nakaseomyces glabrataa.
Minor concerns
- Introduction part is not enough. First two paragraphs in discussion can be used in the introduction instead. And the discussion part should be added with the explanation of the discrepancy in some results between disc diffusion and broth microdilution assays. Additionally, please discuss about the possible mechanism that the Sa-HFB1 can function as the antifungal protein, including explanation about why you have to study and report result involving derivatives.
- Spelling, punctuation, italic names of microorganisms. They all have to be corrected wherever they presented in the manuscript. I will mention some of them as follows.
- In 2.2, the formula for alkaline medium should be better to present in form of molar concentrations. What is 15o Balling means? And should it be stationary conditions in line 68?
- In 2.4, PCR programming should use the word “for” instead of symbol hyphen.
- After 2.2, should follow with 2.5
- Some topics capitalized the first letter in each word, some are not. Please make them same pattern.
- Should MALDI-TOF Ms be MS? Please make them consistent whichever you choose.
- Many figures have low visuality. I cannot see them clearly. Fig. 3b, figure 9 for examples. Some of them look like the captured picture. Please do some processing to make them more visible.
- Description for the full name of drugs in table 1 and 2 are needed.
- Reference part. Please check the pattern in consistency. I saw some title of the articles capitalized the first letter of each word; some are not. Reference 16 is not necessary. This website can be added in the bracket in the material and method part where it should be mentioned.
Author Response
Dear Reviewer,
Thank you so much for your revision. All data included in revision version of the manuscript.
This study involved purification and characterization of the hydrophobin isolated from the alkaliphilic fungus Sodionyces alkalinus. The result of study showed novelty and very interesting to the medical area. There are some concerning issues as followings.
Major concerns
- The antifungal susceptibility testing should be done again. The results in table 2 is not completed. C. neoformans297m had not tested with the conventional drugs so there was no control for this organism. I also concerned about inconsistent results between disc diffusion and broth dilution method for C.albicans 1585m. The disc diffusion method showed that Sa-HFB1 has larger zone, but it showed equivalent MIC. Would it be possible if the discrepancy came from improper techniques in the disc diffusion assay. As I saw in figure S2, the tested organisms were not spread entirely on the surface of the medium. This low inoculum size could affect the killing ability of diffused drugs. Please consider re-performing this experiment.
The antifungal tests of the C. neoformans 297m with the conventional drugs and photo of the disc diffusion assay were added in the figure 2 section Supplementary and discussed in the result parts in the manuscript.
- Several names of the fungi included in this study need to be changes. According to a paper titled “Name changes for fungi of medical important 2018 to 2019. J Clin Microbiol 2021. 59(2): e01811-20. The name of Candida krusei and C. glabrata were changed to Pichia kudriavzevii and Nakaseomyces glabrataa.
Thank you for pointing these moments! Name of all fungal species have been changed to new taxa.
Minor concerns
- Introduction part is not enough. First two paragraphs in discussion can be used in the introduction instead. And the discussion part should be added with the explanation of the discrepancy in some results between disc diffusion and broth microdilution assays. Additionally, please discuss about the possible mechanism that the Sa-HFB1 can function as the antifungal protein, including explanation about why you have to study and report result involving derivatives.
We have revised introduction part according to your comments and established a new submission.
- Spelling, punctuation, italic names of microorganisms. They all have to be corrected wherever they presented in the manuscript. I will mention some of them as follows.
- In 2.2, the formula for alkaline medium should be better to present in form of molar concentrations. What is 15o Balling means? And should it be stationary conditions in line 68?
- In 2.4, PCR programming should use the word “for” instead of symbol hyphen.
- After 2.2, should follow with 2.5
Thank you so much, that you noted these errors in the text. We excluded “stationary conditions” from the text. All punctuation, italic names of microorganisms have been corrected throughout the revised manuscript.
- Some topics capitalized the first letter in each word, some are not. Please make them same pattern.
- Should MALDI-TOF Ms be MS? Please make them consistent whichever you choose.
- Many figures have low visuality. I cannot see them clearly. Fig. 3b, figure 9 for examples. Some of them look like the captured picture. Please do some processing to make them more visible.
Description for the full name of drugs in table 1 and 2 are needed.
We included all necessary information in the table 1 and 2. We corrected the Fig3B and Fig 9.
- Reference part. Please check the pattern in consistency. I saw some title of the articles capitalized the first letter of each word; some are not. Reference 16 is not necessary. This website can be added in the bracket in the material and method part where it should be mentioned.
Reference part in the manuscript have been corrected
Reviewer 2 Report
This manuscript is a contribution to the Special Issue "Fungi and Fungal Metabolites for the Improvement of Human and Animal Life, Nutrition and Health 2.0". The aim of this Special Issue is to “…to encourage authors working in this field to publish their most recent work in this rapidly growing journal in order for the large readership to appreciate the full potential of wonderful and beneficial fungi.”
The authors of this manuscript would benefit from referring to the papers already published in this same Special Issue (https://www.mdpi.com/journal/jof/special_issues/Fungi_Metabolites). Significant revisions are needed to bring this article up to the standard required for submission to a Special Issue.
Overall, this manuscript would lend itself better to a journal oriented towards chemistry rather than micro/biology, as the emphasis of this report is on discovering the identity of the native compound and its characterisation. There is minimal effort to assess its antifungal activity and effects. There is very little mycology in this paper and what is included falls short of the minimum standards used for the determination of minimal inhibitory concentrations. For example, the quality of the data in the table that express inhibition zones is questionable according to the images provided in the Supplementary materials as the zones are not circular and there is evidence of trailing endpoints (as observed with triazoles); neither of these anomalies is discussed. Moreover, the table demonstrating MICs is incomplete and uses incorrect units. For a Journal of Fungi paper, there should be testing and evidence of efficacy against many more and varied clinical isolates.
The main problem with this manuscript is a lack of detail in how the hydrophobin F11 was chosen as a candidate antifungal compound. Where is the evidence that suggests this molecule is worthy of this investigation? This evidence must be cited in the introduction. On further reading, this information incorrectly appears in the discussion. Considering this hydrophobin was already identified, there should be equal emphasis on confirming and characterising the structure AND demonstrating its antifungicidal effects using a wider breadth of fungal isolates, yet the latter is lacking from the article.
Another critical fault of this manuscript is the quality of English. Review by an English-speaking native would vastly improve the comprehension.
Author Response
Dear Reviewer,
Thank you so much for your revision. All data included in revision version of the manuscript.
This manuscript is a contribution to the Special Issue "Fungi and Fungal Metabolites for the Improvement of Human and Animal Life, Nutrition and Health 2.0". The aim of this Special Issue is to “…to encourage authors working in this field to publish their most recent work in this rapidly growing journal in order for the large readership to appreciate the full potential of wonderful and beneficial fungi.”
The authors of this manuscript would benefit from referring to the papers already published in this same Special Issue (https://www.mdpi.com/journal/jof/special_issues/Fungi_Metabolites). Significant revisions are needed to bring this article up to the standard required for submission to a Special Issue.
The paper of Deshmukh, S.K.; Dufossé, L.; Chhipa, H.; Saxena, S.; Mahajan, G.B.; Gupta, M.K. Fungal Endophytes: A Potential Source of Antibacterial Compounds. J. Fungi 2022, 8, 164. https://doi.org/10.3390/jof8020164 from the Special Issue is referenced in the relevant part of the Introduction.
Overall, this manuscript would lend itself better to a journal oriented towards chemistry rather than micro/biology, as the emphasis of this report is on discovering the identity of the native compound and its characterisation. There is minimal effort to assess its antifungal activity and effects. There is very little mycology in this paper and what is included falls short of the minimum standards used for the determination of minimal inhibitory concentrations. For example, the quality of the data in the table that express inhibition zones is questionable according to the images provided in the Supplementary materials as the zones are not circular and there is evidence of trailing endpoints (as observed with triazoles); neither of these anomalies is discussed. Moreover, the table demonstrating MICs is incomplete and uses incorrect units. For a Journal of Fungi paper, there should be testing and evidence of efficacy against many more and varied clinical isolates.
Thank you for your recommendation! Two other opportunistic fungal tests were added in the manuscript in the Table 1 in the Result section. The antifungal tests of the C. neoformans 297m and Aspergillus fumigatus 390m photo of the disc diffusion assay were added in the Supplementary materials.
The main problem with this manuscript is a lack of detail in how the hydrophobin F11 was chosen as a candidate antifungal compound. Where is the evidence that suggests this molecule is worthy of this investigation? This evidence must be cited in the introduction. On further reading, this information incorrectly appears in the discussion. Considering this hydrophobin was already identified, there should be equal emphasis on confirming and characterising the structure AND demonstrating its antifungicidal effects using a wider breadth of fungal isolates, yet the latter is lacking from the article.
Thank you. We rewrote it. The role of hydrophobins in fungal biology has been added in the Introduction part. Active fraction of the ethyl acetate extract was identified (by molecular weight, absorption ratio at certain wavelengths methods) as an antimicrobial peptide and this is the first time for separation of this compound from Sodiomyces alkalinus and was shown to exert potent antifungal activities. We suppose that the antifungal activity of Sodiomyces alkalinus could be explained by synthesis of this compound. This is the first time that the antifungal activity of hydrophobin from any fungi has been reported. We also included minor comments in the Results section. Also, some new fungal isolates tests were added in table 2.
Another critical fault of this manuscript is the quality of English. Review by an English-speaking native would vastly improve the comprehension.
Thank you so much, that you noted these grammar errors in the text. This manuscript was proofreaded by MDPI service.
Round 2
Reviewer 1 Report
The major revision on antifungal testing has been improved in the manuscript. However, there are several points of minor mistakes that the authors need to take more intension to make the corrections.
1. Fungal name change of Candida. The changed name were switched. This mistake appears both in result text and table 1. In addition, the old names still appear in some places in result and discussion.
2. Figures: Most of the picture's visibility are still poor (fig 2-8). Figure 1 is too large. Figure 9 is out of page.
3. MIC unit should be per mL.
4. Reading time of MIC for C. neoformans, should it be 48 h according to the standard methodology?
5. Line 461, p14, name of fungus isn't italicized.
6. Reference pattern still has to be rechecked. For example, I can see ref. 3 contains pp. for page number while ref 4,6 has only one p and most references have nothing.
7. In table 1, is compound 40 mg/ml corrected? Also, the information in last row of the table has to be aligned.
Author Response
1. Fungal name change of Candida. The changed name were switched. This mistake appears both in result text and table 1. In addition, the old names still appear in some places in result and discussion.
All required changes have been made to the manuscript.
2. Figures: Most of the picture's visibility are still poor (fig 2-8). Figure 1 is too large. Figure 9 is out of page.
The figures have been changed
3. MIC unit should be per mL.
MIC unit have been changed per µg/mL
4. Reading time of MIC for C. neoformans, should it be 48 h according to the standard methodology?
Incubation for 48 h gave adequate growth with low standard deviations, and 48 h was selected as the optimal incubation period for this study.
5. Line 461, p14, name of fungus isn't italicized.
We have changed it according to the comments
6. Reference pattern still has to be rechecked. For example, I can see ref. 3 contains pp. for page number while ref 4,6 has only one p and most references have nothing.
We have changed it according to the comments
7. In table 1, is compound 40 mg/ml corrected? Also, the information in last row of the table has to be aligned.
We have changed it
Reviewer 2 Report
The authors have made some minor amendments to the manuscript but not addressed some of this reviewer's concerns about the quality of the susceptibility testing (concentrations are not standard for reporting MICs) and discussion, presentation of the figures, and English language; for example, many spelling mistakes are retained and the Supplementary figures are still labelled incorrectly and lack detailed legends. This reviewer’s comment to the authors about referring to another publication in the Special Issue was misunderstood – the authors should review the quality of these publications to improve their submission, not cite a publication (unless relevant). This reviewer has attached some editorial comments that are by no means complete since there were so many revisions needed. However, this reviewer recommends that this list is addressed to improve the article.

Author Response
This reviewer recommends that this list is addressed to improve the article.
We have adrested all the questions raised by the reviewer. The authors would like to express sincere gratitude to the aninymous referee for the scrutiny and careful reading of the manuscript that has allowed to greatly improve the exposition.